# Transcriptomic and Proteomic Profiles for Elucidating Cisplatin Resistance in Head-and-Neck Squamous Cell Carcinoma

**DOI:** 10.3390/cancers14225511

**Published:** 2022-11-09

**Authors:** Yoelsis Garcia-Mayea, Lisandra Benítez-Álvarez, Almudena Sánchez-García, Marina Bataller, Osmel Companioni, Cristina Mir, Sergi Benavente, Juan Lorente, Nuria Canela, Ceres Fernández-Rozadilla, Angel Carracedo, Matilde E. LLeonart

**Affiliations:** 1Biomedical Research in Cancer Stem Cells Group, Vall d’Hebron Research Institute (VHIR), 08035 Barcelona, Spain; 2Departament de Genètica, Microbiologia i Estadística, Universitat de Barcelona, 08028 Barcelona, Spain; 3Department of Biochemistry, Massey Cancer Center, Virginia Commonwealth University, Richmond, VA 23298, USA; 4Eurecat Centre Tecnològic de Catalunya-Centre for Omic Sciences (COS), Joint Unit University of Rovira i Virgili-EURECAT, 43204 Reus, Spain; 5Instituto de Investigación Sanitaria de Santiago de Compostela (IDIS), 15706 Santiago de Compostela, Spain; 6Fundación de Medicina Xenómica (SERGAS), 15706 Santiago de Compostela, Spain; 7Grupo de Medicina Xenómica, Centro de Investigación de Red de Enfermedades Raras (CIBERER), CIMUS, University of Santiago de Compostela, 15706 Santiago de Compostela, Spain; 8Spanish Biomedical Research Network Centre in Oncology, CIBERONC, Vall d’Hebron Research Institute (VHIR), Passeig Vall d´Hebron 119–129, 08035 Barcelona, Spain

**Keywords:** cancer, HNSCC, chemotherapy resistance, cisplatin, RNA-seq, proteomics

## Abstract

**Simple Summary:**

Most treatment failures in head and neck squamous cell carcinoma (HNSCC) patients are due to the presence of resistant cells. Despite the chemotherapeutic advances that have taken place in recent decades, there are hardly new alternatives for HNSCC patients, with cisplatin being the most widely used chemotherapy drug. Therefore, it is urgent to propose new potential biomarkers and alternative therapeutic modalities capable of preventing the acquisition of resistance to treatment. We have conducted a RNA sequencing and proteomics study in cisplatin resistant and sensitive HNSCC cells with the aim of unravelling the molecular mechanisms involved in chemoresistance. Then, by an extensive literature search, in silico studies in cBioportal and in vitro experiments in biopsies from resistant and sensitive patients, we have listed potentially involved genes and proteins. Overall, the overexpression of *MAGEB2* was identified in resistant tumours, revealing it as a novel protein targeting sensitised HNSCC resistant patients.

**Abstract:**

To identify the novel genes involved in chemoresistance in head and neck squamous cell carcinoma (HNSCC), we explored the expression profiles of the following cisplatin (CDDP) resistant (R) versus parental (sensitive) cell lines by RNA-sequencing (RNA-seq): JHU029, HTB-43 and CCL-138. Using the parental condition as a control, 30 upregulated and 85 downregulated genes were identified for JHU029-R cells; 263 upregulated and 392 downregulated genes for HTB-43-R cells, and 154 upregulated and 68 downregulated genes for CCL-138-R cells. Moreover, we crossed-checked the RNA-seq results with the proteomic profiles of HTB-43-R (versus HTB-43) and CCL-138-R (versus CCL-138) cell lines. For the HTB-43-R cells, 21 upregulated and 72 downregulated targets overlapped between the proteomic and transcriptomic data; whereas in CCL-138-R cells, four upregulated and three downregulated targets matched. Following an extensive literature search, six genes from the RNA-seq (*CLDN1*, *MAGEB2*, *CD24*, *CEACAM6*, *IL1B* and *ISG15*) and six genes from the RNA-seq and proteomics crossover (*AKR1C3*, *TNFAIP2*, *RAB7A*, *LGALS3BP*, *PSCA* and *SSRP1*) were selected to be studied by qRT-PCR in 11 HNSCC patients: six resistant and five sensitive to conventional therapy. Interestingly, the high *MAGEB2* expression was associated with resistant tumours and is revealed as a novel target to sensitise resistant cells to therapy in HNSCC patients.

## 1. Introduction

Head and neck squamous cell carcinoma (HNSCC) is among the top ten cancers, in terms of incidence worldwide, and portends a high lethality [1]. Standard treatment options for the locally advanced disease, the most common form of presentation, include surgery and postoperative radiotherapy with or without cisplatin (CDDP)-based chemotherapy; or radical chemoradiation, based on CDDP, carboplatin plus 5-fluorouracil or cetuximab. However, the local control and overall survival at five years is only 50% with a better prognosis in HPV-related oropharyngeal cancer. Options in the recurrent/metastatic setting were historically contingent on the EXTREME trial (platinum-based chemotherapy plus cetuximab) which achieved a median overall survival of only 7–10 months [2]. In contrast, one of the most significant advances in the treatment of HNSCC has been the use of immunotherapy in patients with recurrent/metastatic cancer, as exemplified by the Checkmate 141 and Keynote 048 trials. Results from the Checkmate 141 trial indicate that nivolumab provides an improved overall survival versus the standard single-agent therapy in platinum-refractory recurrent HNSCC (six-month progression-free survival, 19.7% versus 9.9%) [3], while the Keynote 048 trial shows that pembrolizumab plus platinum-based chemotherapy was superior, in terms of overall survival versus EXTREME in the first-line setting (median overall survival, 13 months versus 10.7 months) [4]. This relative success of immunotherapy in the recurrent/metastatic setting drives active research to incorporate immunotherapy in the curative setting. The late diagnosis in a considerable percentage of patients is directly related to aggressiveness, which is due to a high tumour heterogeneity and the presence of resistant cells [5]. In fact, the treatment success in the curative setting is frequently masked by its resistance to radio- and/or chemotherapy, particularly CDDP. Importantly, the resistance phenotype can be reversed [6,7,8]. Therefore, the therapeutic intervention against those mechanisms in resistant cells is urgently needed to sensitise HNSCC cells to CDDP treatment.

Moreover, techniques, such as whole-exome sequencing and RNA-sequencing (RNA-seq), represent a step forward in personalized medicine [9,10,11]. Cancer transcriptome sequencing captures both coding and noncoding RNA and provides the strand orientation for a complete view of expression dynamics. To detect novel genes related to resistance in the HNSCC model, we performed RNA-seq in CDDP resistant cells previously generated in our laboratory after exposure to CDDP for several months (JHU029-R, HTB-43-R and CCL-138-R) [12,13]. Moreover, we evaluated the proteomic profile of resistant versus sensitive cells of the following cell lines: HTB-43 and HTB-43R; and CCL-138 and CCL-138-R. The results from the proteomic profile of CCL-138 and CCL-138-R have been described [12]. Therefore, in the present study, apart from the RNA-seq studies, we expanded our effort in identifying targets commonly deregulated at the RNA and protein levels in the cell lines HTB-43 and CCL-138. The impact of genes and proteins resulting from the RNA-seq and proteomic analyses was explored by searching the literature, studies in the cBioPortal database and in a group of patients who showed resistance or sensitivity to conventional radio and chemotherapy treatments. Of importance, the MAGE family member B2 (*MAGEB2*) was found to be significantly upregulated in a group of HNSCC patients characterised by their lack of response to conventional treatment (CDDP and/or radiotherapy). Overall, our data identify novel genes deregulated in CDDP resistance that deserve further attention in cancer. These genes may represent potential targets to overcome chemotherapy resistance in HNSCC and possibly other cancers.

## 2. Materials and Methods

### 2.1. Patients

Eleven primary tumours from laryngeal cancer patients were collected from Vall d’Hebron University Hospital (HUVH) for RNA extraction. The study was performed according to the Declaration of Helsinki and approved by the Ethics Committee of Vall d’Hebron Hospital (CEIC) (Ref. PR(AG)342/2016). The informed consent was obtained from all patients. The characteristics of the patients in relation to the treatment received, according to the clinical protocols, are summarized in Appendix A. The clinical follow-up of patients was at least three years from the date of diagnosis.

### 2.2. Cell Culture

The HTB-43 (pharyngeal) and CCL-138 (metastatic pharyngeal) cell lines were obtained from American Type Culture Collection (ATCC, Manassas, VA, USA), and JHU029 (larynx) cells were kindly provided by Dr Aznar-Benitah (IRB). These cell lines were authenticated and cultured in MEM (Gibco, Thermo Fisher Scientific, Waltham, MA, USA) supplemented with 10% fetal bovine serum (FBS) (Biowest, Nuaillé, France) and 1% penicillin–streptomycin (Pen 20 U/mL and Strep 20 µg/mL, Gibco, Thermo Fisher, Waltham, MA, USA). Resistant cell lines were generated as described [12].

### 2.3. Transfection of the Small Interfering RNAs (siRNAs)

The cells were transfected following the reverse transfection protocol, as previously described [13]. The siRNAs used were ones targeting the *MAGEB2* gene and another as a control siRNA (Integrated DNA Technologies), both at a 10 nM concentration.

### 2.4. CDDP Assay

To determine the IC50 of the cells transfected with siRNA *MAGEB2*, a CDDP assay was performed, as previously described [13]. Survival curves and IC50 calculations were performed using GraphPad Prism (GraphPad Software, San Diego, CA, USA).

### 2.5. RNA Extraction

The RNA extraction was performed using the mirVana miRNA Isolation Kit (Ambion, Austin, TX, USA), and RNA was treated with DNAse I, using the DNA-free DNA Removal Kit (Invitrogen, Thermo Fisher Scientific, Waltham, MA, USA), following the manufacturer’s protocols. The RNA quality was analysed by the RNA integrity number (RIN) values with an Agilent 2100 Bioanalyzer system.

### 2.6. RNA-Sequencing (RNA-Seq)

Duplicate samples of the following cell lines: HTB-43, HTB-43-R, CCL-138 and CCL-138-R were used for the RNA-seq studies. One of the samples from the CCL-138 line was unsuitable for the RNA-seq analysis (Appendix A). TruSeq Stranded Total RNA libraries were constructed by Macrogen Inc. (Macrogen Europe, Amsterdam, The Netherlands) to obtain Illumina paired-end reads (151 bp read length). The raw data generated for this study have been deposited in the NCBI-SRA repository under the BioProject accession code PRJNA893841 (Appendix A).

### 2.7. Bioinformatics Workflow for the Differentially Expressed Gene Identification

The detailed bioinformatics workflow used here is available at https://github.com/lisy87/Differential_Expression, accessed on 10 October 2022. Briefly, the Fastq raw data files were analysed with FastQC software v0.11.9 [14] using default parameters, and raw reads were filtered with Trimmomatic v0.39 [15] to remove the low-quality reads and Illumina adapters. The filtered reads were mapped against the GRCh38 reference genome (primary assembly and annotation) with STAR v2.7.8a [16]. The BAM files were filtered with samtools v1.7 to retain the mapped reads with quality scores higher than 40. The read counts were obtained for all samples from the BAM files using the featureCounts tool v2.6.3 with the default options for paired-end reads. We performed a differential expression analysis using the DESeq2 package v1.30.1 [17] in R v4.04 and the parental condition as the denominator of the comparison. The significance was set at a false discovery rate (FDR)-adjusted *p*-value < 0.05 and log2-fold change (log2FC) values of 1. The batch retrieval of the differentially expressed genes was performed using the org.Hs.eg.db package v3.12.0. The gene ontology enrichment analysis of the differentially expressed gene identification was carried out using the goseq library v1.42.0 [18]. Finally, simple bash commands were used to cross the results of the differential expression and the proteomic analysis, as well as the previously described genes.

### 2.8. Proteomic Analysis

Briefly, proteins were analysed on a nano-LC-LTQ-Orbitrap Velos Pro mass spectrometer (Thermo Fisher). The raw data files obtained were analysed by Multidimensional Protein Identification Technology (MudPIT) on Proteome Discoverer software v.1.4.0.288 (Thermo Fisher Scientific). For the protein identification, all MS and MS/MS spectra were analysed using the Mascot search engine (version 2.5). The FDR and protein probabilities were calculated by the Percolator software. For the protein quantification, the ratios between each TMT label against the 126-TMT label were used, and the quantification results were normalized, based on the protein median to reduce the experimental bias and log2 transformed and mean centred for the variance stabilization, data range compression and to make the data more normally distributed before the statistical analysis. The details of the methodology have been described previously [13]. The raw data from the proteomic study are in the repository ProteomeXchange Consortium via the PRIDE [19] partner repository with the dataset identifiers PXD020159 and PXD037118.

### 2.9. qRT-PCR

cDNA was obtained using the RevertAid H Minus First Strand cDNA Synthesis Kit (Thermo Fisher Scientific, Waltham, MA, USA) from the RNA extraction, both cell lines and patient samples. The expression of the following genes was analysed in triplicate by a quantitative real-time PCR (qRT-PCR), using Taqman probes (Life Technologies): *CD24* (Hs02379687_s1); *CEACAM6* (Hs03645554_m1); *CLDN1* (Hs00221623_m1); *MAGEB2* (Hs01890983_s1); *IL1B* (Hs01555410_m1); *RAB7A* (Hs01115139_m1); *LGALS3B* (Hs00174774_m1); *SSRP1* (Hs00172629_m1); *ISG15* (Hs01921425_s1); *PSCA* (Hs04177224_g1); *AKR1C3* (Hs00366267_m1) and *TNFA1P2* (Hs00969305_m1). *IPO8* (Hs00183533_m1) and *TBP* (Hs00427620_m1) were used as endogenous genes. LightCycler 480 multiwell plates 384 (#04729749001, Life Science) were used to carry out the qRT-PCR with the following amounts per reaction: 3.25 μL of nuclease-free water; 5 μL of TaqMan Universal Master mix II no UNG (#4440040, Applied Biosystems, Life Technologies); 0.5 μL of TaqMan probe; and 1.25 μL of cDNA. The real-time PCR was performed on a LightCycler 480 machine (LifeScience), and the quantification method used for all assays was 2-ΔΔCt. ΔΔCt = [(CT of the gene of interest − CT of the endogenous control) experimental condition − (CT of the gene of interest − CT of the endogenous control) control condition].

### 2.10. Clinical Data Analysis (In Silico Study)

In order to analyse whether the upregulated genes in resistant cell lines are correlated with lower survival rates of patients, the TCGA data was used. The curated nonredundant set of TCGA studies for HNSCC, with mutation, structural variant and copy number alteration data, including 14 studies with 1932 patients, was selected from cBioPortal. Then, this composite dataset was queried for different datasets; the altered and unaltered samples were downloaded, which were custom selected in the combined study, and groups were created for comparison of the altered and unaltered genes. In the case of a sufficient sample size, the data were filtered for tumour location (pharynx, larynx, oral cavity, nose) analysis. Oncoprint graph and survival curves were recorded. Samples were excluded when they overlapped between groups. Survival curves were computed between the tumour samples that had at least one alteration in one of the query genes and tumour samples that did not. The cBioportal results are displayed as Kaplan–Meier plots with significant *p*-values and *q*-values from a log rank test. Some genes are present in more than one dataset; in that case, the single gene survival curves are reported only once. Additionally, survival curves for all genes together were estimated for all datasets.

### 2.11. Statistical Analysis

For the RNA-seq analyses, we considered a limited fold change (LFC) = 1 and *p*-value = 0.05 for the duplicate samples. The statistical analysis to find the significant protein changes between the conditions included in the present studies was performed on Mass Profiler Professional software v.14.5 from Agilent Technologies. A one-way ANOVA for all cell groups and an unpaired Student’s *t*-test for the CDDP resistant versus parental cells, were analysed in CCL-138 and HTB-43 cells. In both cases, a Benjamini–Hochberg *p*-value correction for the multiple comparisons was applied to reduce the false-positive findings. In these studies, only those proteins that were quantified in at least 3 independent experiments were considered for statistical purposes. For the qRT-PCR studies, a Student’s *t*-test was performed to compare the differences between the two groups. All tests were two-tailed. A *p*-value < 0.05, 0.01 or 0.001 (indicated in the plots as *, ** and ***, respectively) was considered significant. cBioportal (TCGA) results are displayed as Kaplan–Meier plots with significant *p*-values and *q*-values from a log rank test. Some genes are present in more than one dataset; in that case, the single gene survival curves are reported only once. Additionally, survival curves for all genes together were estimated for all datasets. The Pearson test determined the association of several genes with the tumour stage.

## 3. Results

### 3.1. RNA-Seq Reveals Some Potential Targets of CDDP Resistance in the HNSCC Cell Lines

The following cell lines were analysed by RNA-seq: JHU029, JHU029-R, HTB-43, HTB-43-R, CCL-138 and CCL-138-R. The heatmap graphs showed the gene expression profiles of the parental and resistant cells quite well (Appendix A). For the JHU029-R cells, the results indicated 29 upregulated genes and 84 downregulated genes (Figure 1A–C, upper panel and Appendix A). In the HTB-43-R cells, 263 genes were upregulated and 392 were downregulated (Appendix A); in the CCL-138-R cells, 154 genes were upregulated and 67 were downregulated (Appendix A). To select a few genes to validate by the qRT-PCR in the respective HNSCC cell lines, based upon the literature, potential key genes with crucial involvement in HNSCC tumours were selected from our RNA-seq list [20,21,22,23,24,25]. In particular, we focused on the upregulated genes to identify future cancer targets. The genes of interest were as follows: JHU029-R cells, claudin 1 (*CLDN1*) and MAGE family member B2 (*MAGEB2*); HTB-43-R cells, signal transducer CD24 (*CD24*) and CEA cell adhesion molecule 6 (*CEACAM6*); and CCL-138-R cells, interleukin 1 beta (*IL1B*) and ubiquitin-like protein ISG15 (*ISG15*). The rates of alteration for these genes, according to TCGA, were 13% for *CLDN1*, 2.2% for *MAGEB2*, 1.2% for *ISG15*, 1% for *IL1B*, 0.6% for *CEACAM6* and 0% for *CD24* (Appendix A). These six genes were evaluated by the qRT-PCR, in triplicate, in the following cells: JHU029, JHU029-R, HTB-43, HTB-43-R, CCL-138 and CCL-138-R. The genes *CLDN1* and *MAGEB2* were upregulated in the JHU029-R cells, as shown by RNA-seq (Figure 1A); note that *MAGEB2* was not expressed in the HTB-43 and CCL138 cell lines. Figure 1B (lower panel) shows that *CD24* and *CEACAM6* were upregulated in the HTB-43-R cells. Figure 1C (lower panel) shows that *IL1B* and *I5G15* were upregulated in the CCL-138-R cells. Moreover, *IL1B* and *I5G15* were also upregulated in the JHU029-R cells (Figure 1A, lower panel). According to the in silico study (cBioPortal), only the *CLDN1* gene gave significant results in relation to the survival of HNSCC patients. Patient survival was lower in cases where the *CLDN1* gene is altered (45.93 [34.07–76.18]) versus those where it is not altered (65.77 [57.47–78.70]) (*p* = 0.021) (Figure 1D). Of interest, most of the proteins corresponding to the deregulated genes altered in resistant versus sensitive lines, belonged to the group of metabolic interconversion enzymes (Appendix A).

### 3.2. Proteomic Study in HTB-43 and CCL-138 Cell Lines in Relation to the RNA-Seq Results

The CDDP resistant versus parental CCL-138 and HTB-43 cells were analysed by proteomics. The results of the identified and deregulated proteins in the HTB-43 cells are shown (Appendix A). The study of the commonly deregulated proteins between HTB-43 and CCL-138 is shown (Appendix A). In particular, in the case of the CCL-138 cells, two overexpressed proteins and their direct association with CDDP resistance have been fully characterized in our laboratory [13,26].

Moreover, crucial genes involved in resistance, according to the literature were found to be deregulated in both lists of the RNA-seq and deregulated proteins (Appendix A) [27,28,29]. This is the case for some types of ABC transporters, such as *ABCA1* and *ABCC1*, and other resistance-related proteins, such as *AKR1C1*, *HIF-1α* and *STAT3*.

### 3.3. Combinatorial Results of the Altered Genes and Proteins in the HTB-43 and CCL-138 Cell Lines

The gene expression profile and the proteomic profile of the HTB-43, HTB-43-R, CCL-138 and CCL-138-R cells were compared. The intersection of the gene expression and proteomics data revealed that 21 genes/proteins were upregulated, and 72 genes/proteins were downregulated in the HTB-43-R cells versus the parental cells (Figure 2A). For the CCL-138-R cells, a total of four genes/proteins were upregulated, and three were downregulated, compared with parental cells (Figure 2B). Following a review of the literature, a reduced number of genes with a potential involvement in HNSCC tumours were finally selected for study through the TCGA database [30,31,32,33,34,35,36,37,38,39,40]. The number of genes was reduced to the following: prostate stem cell antigen (*PSCA*), member of the RAS oncogene family (*RAB7A*), TNF alpha induced protein 2 (*TNFAIP2*), glycoprotein nmb (*GPNMB*), niban apoptosis regulator 2 (*NIBAN2*), structure specific recognition protein 1 (*SSRP1*), aldo-keto reductase family 1 member C3 (*AKR1C3*), lipocalin 2 (*LCN2*), sperm associated antigen 9 (*SPAG9*), galectin 3 binding protein (*LGALS3BP*) and eukaryotic translation elongation factor 1 alpha 2 (*EEF1A2*) (Figure 2C). Alterations of these genes were evident in 8% of cases for *PSCA*, 3% for *RAB7A*, 1.8% for *TNFAIP2*, 1.5% for *GPNMB*, 1.4% for *NIBAN2*, 1.4% for *SSRP1*, 1.3% for *AKR1C3*, 1.2% for *LCN2*, 0.9% for *SPAG9*, 0.9% for *LGALS3BP* and 0.8% for *EEF1A2* (Appendix A). In silico studies showed that for all genes independently or together, the HNSCC patient survival was not significantly different in the group of patients with altered genes, compared to those with unaltered genes.

### 3.4. Study of the Genes Related to Resistance in Laryngeal Cancer Patients

A group of 11 patients were selected from the Hospital Vall d’Hebron, according to their response to CDDP and/or radiotherapy (Appendix A). All patients were diagnosed at a similar stage and treated under similar circumstances with radiotherapy and/or CDDP after surgery. Six were resistant patients, as radiotherapy and/or CDDP therapy was not effective for them. In contrast, five patients responded well to radiotherapy and/or CDDP treatment, showing tumour remission and currently free of the disease three years after diagnosis. Normal and tumour tissues from such patients were analysed by qRT-PCR for the following 12 genes; six from RNA-seq: *CLDN1*, *MAGEB2*, *CD24*, *CEACAM6*, *IL1B* and *ISG15* and six from RNA-seq and proteomics cross-linking: *AKR1C3*, *TNFAIP2*, *RAB7A*, *LGALS3BP*, *PSCA* and *SSRP1*. The endogenous genes *TBP* and *IPO8* were included, as described [41] (Figure 3A). The only gene able to distinguish the resistant versus sensitive group was *MAGEB2* (Figure 3B,C). However, the *PSCA* expression showed a trend towards significance (Appendix A). Moreover, in silico studies showed that alterations in the set of these 12 genes correlated with the tumour stage in a total of 897 HNSCC patients (Figure 3D). Note that those patients with concomitant alterations in these 12 genes are mostly at the T4 category, a clinical parameter that refers to the most advanced and invasive type of primary tumours. The survival of patients with changes in all of these genes, alone or together, was not significantly different from those with unaltered genes.

### 3.5. Study of MAGEB2 in the Sensitisation of the Resistant Cells

To determine whether the *MAGEB2* inhibition was able to sensitize resistant cells, we transfected the JHU029-R cells with a siRNA against *MAGEB2*. The ability of the siRNA against *MAGEB2* to inhibit the *MAGEB2* gene expression was approximately 30-fold (Figure 3E). Inhibition of *MAGEB2* was able to sensitize the JHU029-R cells to the effect of the CDDP treatment (Figure 3F). In contrast, the CCL-138 cells that did not express *MAGEB2* (Figure 1C) showed no sensitization effect after the *MAGEB2* depletion, as expected (Appendix A). Potential interacting proteins with MAGEB2 are shown in Appendix A.

## 4. Discussion

CDDP is the ideal chemotherapy for a myriad of cancer types. However, resistance to CDDP has become a major obstacle in treating HNSCC patients. The following known hallmarks of HNSCC have been described: *TP53*, *EGFR* and *K-ras* mutations; TGF-α, VEGF, PDGF, FGF-1, PI3K, AKT and MDM2 protein overexpression; and PTEN, pRb and NOTCH-1 downregulation [42,43,44]. However, despite the number of proposed predictive markers, none are currently in clinical use or validated in a large cohort; therefore, there are no informative markers, in terms of predicting a possible CDDP response [45].

Moreover, in HNSCC, transcriptomic signatures provide promising results with a potential therapeutic impact [46]. Here, we describe the RNA-seq results from three different HNSCC cell lines, according to their location (laryngeal, pharyngeal and metastatic pharyngeal cells). Interestingly, the transcriptional patterns are very different when comparing the resistant cells versus the sensitive cells. For the JHU029 cells, some deregulated genes belong to the tetraspanin (*TSPAN*) family and the melanoma antigen gene (*MAGE-I*) family of the tumour-specific antigens, whose expression is deregulated in cancer; for the HTB-43 cells, a large number of deregulated genes are associated and interact with the EGFR-associated pathways, and for the CCL-138 cells, some genes belong to the histone family [44,47,48]. Moreover, some known genes clearly associated with resistance, according to the literature, were found to be upregulated in our RNA-seq results from the different cell types, reinforcing the robustness of our data. For example, the genes belonging to the ATP binding cassette family *ABCA1*, *ABCA12*, *ABCA2*, or associated with stemness *ALDH1A3*, *SOX2* and *WNT9A* [49,50]. Of note, the fact that the proteins encoded by the altered genes in resistant versus sensitive lines belong mostly to the group of metabolic interconversion enzymes, highlights the importance of metabolism in the acquisition of the resistant phenotype [51,52].

According to the protein expression panel, a considerable number of proteins are commonly expressed by the HTB-43 cell line (primary pharyngeal cells) and CCL-138 (metastatic pharyngeal cells). This finding could suggest that the identification of these proteins or a group of them in the primary lesion could be useful to potentially predict those patients who will progress in the short-medium term to the metastatic disease. The overlap between genes and proteins commonly deregulated in resistant versus parental HTB-43 and CCL-138 cells, indicates that changes at the transcription level do not always result in protein alterations. While protein alterations have been the focus of therapies, transcriptional changes have been underestimated [53,54,55]. It is surprising that CCL-138 cells showed a very limited number of genes that correlate with the protein expression in comparison with the HTB-43 cells. This finding can be associated with the nature of metastatic CCL-138 cells, where myriad alterations solely at the transcriptional level might have severe consequences on the final phenotype.

Moreover, *CLDN1* (Claudin-1) -the only significant gene from in silico studies-, is a major component of the tight junction molecules that play a role in cell-cell communication and interactions. A recent study proposed the role of *CLDN1* as an EMT regulator in HNSCC [20]. The fact that *CLDN1* represents a prognostic factor in patient survival supports its role as an HNSCC biomarker.

Despite the limitations of our study (small number of patients), the most interesting finding is that the *MAGEB2* expression was revealed to be the only gene able to distinguish resistant versus sensitive laryngeal cancer patients to conventional treatment. Further evidence for the involvement of *MAGEB2* in modulating resistance comes from the fact that its inhibition in resistant laryngeal cancer cells (JHU029-R) sensitises these cells to the effects of CDDP. According to the proteins that interact with MAGEB2, 50% of them belong to the cancer/testis antigen gene (GAGE) family. GAGE proteins are encoded by genes that are normally expressed only in the human germline, but are also expressed in various tumour types. We hypothesise that MAGEB2 could act at the level of two mechanisms of action largely mediated by the GAGE family: (i) ability to create a more “open” chromatin structure by up-regulating histone 3 lysine 56 (H3K56Ac) acetylation, which could increase the efficiency of DNA repair, as described in a model of radioresistance in ovarian cancer or (ii) MAGEB2 could promote the development of immunosuppressive EMT (e.g., by down-regulating the expression of CXCL9 and CXCL10) and decreasing the recruitment of effector CD8+ T cells, thereby exerting resistance to checkpoint immunotherapy. Alternatively, MAGEB2 could induce stemness characteristics by modulating the HDAC1 levels [56,57,58]. The observed oncogenic effect of *MAGEB2* promoting resistance correlates with other described aspects of *MAGEB2* promoting the cell proliferation in tumour cells. For example, it has been reported that *MAGEB2* promotes the cell proliferation in transformed oral keratinocytes, osteosarcoma and colon cancer cell lines [21,59]. Moreover, *MAGEB2* plays a role in enhancing the ribosome biogenesis as part of its repertoire to support cancer cell proliferation and is epigenetically activated in HNSCC [21,60]. Interestingly, the *MAGEB2* expression has been associated with the inflammatory response in tumour laryngeal tissue [57].

Overall, in this study, we revealed new potential therapeutic targets, based on the gene expression (RNA-seq) and proteomic analysis in resistance models. Here, we propose a novel function of *MAGEB2* related to the CDDP resistance in HNSCC tumours, expanding its potential oncogenic role in cancer resistance. The association of the *MAGEB2* expression with the CDDP response suggests its important clinical potential as a target for cancer therapy.

## 5. Conclusions

Our findings demonstrated that omics technologies have a great potential to identify future therapeutic targets. In our case, we have applied these studies in the HNSCC model with the idea of suggesting alternative options for HNSCC patients who do not respond to conventional treatment. RNA-seq and proteomics studies have allowed us to identify 12 genes as potential targets. In particular, the most promising gene resulting from this study is MAGEB2, whose downregulation in vitro induces the sensitisation of cells to CDDP. Importantly, the MAGEB2 expression can distinguish between the CDDP-responsive and CDDP-unresponsive patients.

## Figures and Tables

**Figure 1 cancers-14-05511-f001:**
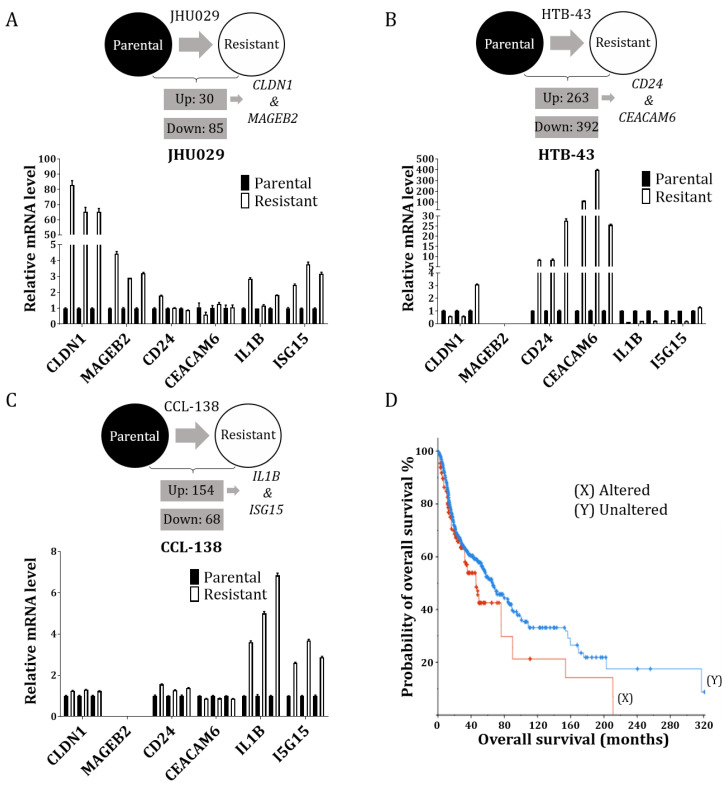
Differentially expressed genes in the resistant versus parental HNSCC cells were identified: (**A**) Thirty genes were found to be upregulated, and 85 were downregulated in the JHU029 cells. An extensive literature search guided us to select two proteins for the qRT-PCR validation: *CLDN1* and *MAGEB2*; (**B**) Two hundred sixty-three genes were upregulated, and 392 were downregulated in the HTB-43 cells. An extensive literature search guided us to select two proteins for the qRT-PCR validation: *CD24* and *CEACAM6*; (**C**) One hundred fifty-four genes were upregulated, and 68 were downregulated in the CCL-138 cells. An extensive literature search guided us to select two proteins for the qRT-PCR validation: *IL1B* and *I5G15*; (**D**) In silico study of the association of *CLDN1* with overall survival. In the curve, each vertical step indicates one or more events (e.g., death), and right-censored patients are indicated by a vertical mark in the curve.

**Figure 2 cancers-14-05511-f002:**
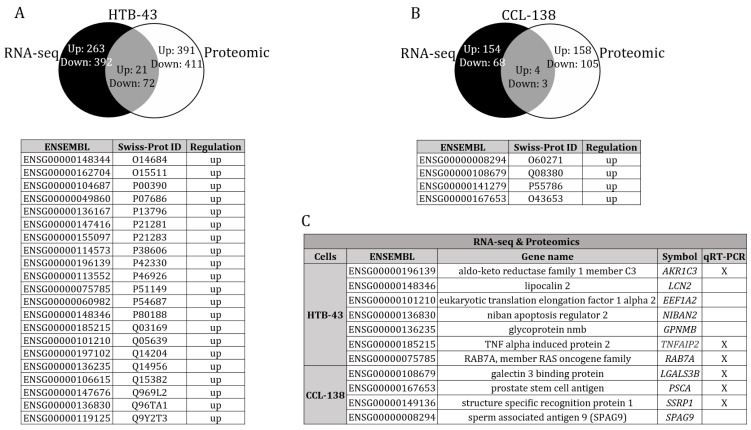
Merge between the differentially expressed genes and the proteins in HNSCC cells: (**A**) A total of 21 proteins were upregulated, and 72 were downregulated in the HTB-43 cells. Note that only the upregulated proteins are shown; (**B**) A total of four upregulated proteins and three downregulated proteins were found in the CCL-138 cells. Note that only the upregulated proteins are shown; (**C**) An extensive literature search guided us to select six.

**Figure 3 cancers-14-05511-f003:**
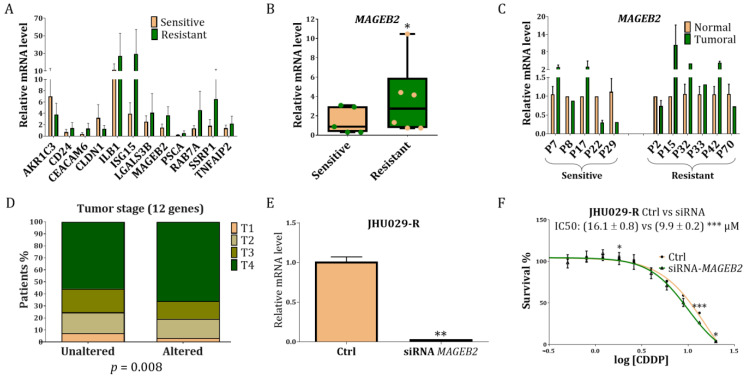
mRNA and survival studies of the selected genes in resistant versus sensitive laryngeal cancer patients: (**A**) qRT-PCR of *CLDN1*, *MAGEB2*, *CD24*, *CEACAM6*, *IL1B*, *I5G15*, *AKR1C3*, *TNFA1B2*, *RAB7A*, *LGALS3B*, *PSCA* and *SSRP1* in both types of patients; (**B**) Significant downregulation of *MAGEB2,* comparing both types of patients in a box plot where each dot is a patient; (**C**) Differences in the *MAGEB2* expression, comparing each patient individually; (**D**) In silico study of the 12 abovementioned selected genes with a tumour stage in HNSCC patients, according to the TCGA dataset; (**E**) Levels of RNA of *MAGEB2* in the transfected cells with a siRNA against *MAGEB2*; (**F**) Sensitizing effect of the JHU029-R cells to CDDP under the *MAGEB2* inhibition. * *p* < 0.05, ** *p* < 0.01, *** *p* < 0.001.

## Data Availability

The data presented in this study are available upon request from the corresponding author. For Appendix A, the raw data is available in the NCBI SRA database, under the following accession numbers and the BioProject accession code PRJNA893841. The raw data from the proteomic study are in the repository ProteomeXchange Consortium via the PRIDE partner repository with the dataset identifiers PXD020159 and PXD037118.

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
