# Peer review of "Transcriptomic and Proteomic Profiles for Elucidating Cisplatin Resistance in Head-and-Neck Squamous Cell Carcinoma"

_cancers, 2022, doi:10.3390/cancers14225511_

Round 1

Reviewer 1 Report

Thank you for submitting an interesting article on looking for reasons for resistance to cisplatin therapy used in HNSCC.

Some corrections / clarifications should be made prior to publication.

1. You write that in 50 years nothing has changed in the treatment of HNSCC in 50 years. I cannot agree with that. The way this cancer is treated has changed over the past few years. Please see the results of the EXTREME, Checkmate-141 and Keynote048 clinical trials and refer to it. Please remember on distinguishing radical (radio/radiochemo) and palliative treatment.   2. page 8 line 298; T refers to tumor and has nothing to do with metastatic leasions according to TNM, please explain.   3. Please write the limitations of study (obviously small number of patients tissue)

Best regards

Author Response

  1. You write that in 50 years nothing has changed in the treatment of HNSCC in 50 years. I cannot agree with that. The way this cancer is treated has changed over the past few years. Please see the results of the EXTREME, Checkmate-141 and Keynote048 clinical trials and refer to it. Please remember on distinguishing radical (radio/radiochemo) and palliative treatment.   

We thank the reviewer for this comment. Our objective was to highlight that overall survival in head and neck cancer is still poor but we do acknowledge that relevant clinical advances have occurred recently. Standard treatment options in the curative and palliative settings as well as landmark immunotherapy trials have been added.  Accordingly, the paragraph in line 55 now reflects those changes and references have been updated.

Line 55: has been changed by the following paragraph (blue colour):

Head and neck squamous cell carcinoma (HNSCC) is among the top ten cancers in terms of incidence worldwide and portends high lethality (1). Standard treatment options for locally advanced disease, the most frequent form of presentation, include surgery and postoperative radiotherapy with or without chemotherapy based on cisplatin; or radical chemoradiation based on cisplatin, carboplatin plus 5-fluorouracil or cetuximab. However, local control and overall survival at 5 years is only ~50% with better prognosis in HPV-related oropharyngeal cancer. Options in the recurrent/metastatic setting were historically contingent on the EXTREME trial (platinum-based chemotherapy plus cetuximab) achieving a median overall survival of only 7-10 months (Nenclares). On the contrary, one of the most significant advances in the treatment of HNSCC has been the use of immunotherapy in patients with recurrent/metastatic cancer as exemplified by the Checkmate 141 and the Keynote 048 trials. Results from the Checkmate 141 trial indicate that nivolumab results in longer overall survival versus standard single-agent therapy in platinum-refractory recurrent HNSCC (progression-free survival at 6 months, 19.7% vs 9.9%) (Ferris), while the Keynote 048 trial shows that pembrolizumab plus platinum-based chemotherapy was superior in overall survival versus EXTREME in the first-line setting (median overall survival, 13 months vs 10.7 months) (Burtness). This relative success of immunotherapy in the recurrent/metastatic setting fuels active investigation to incorporate immunotherapy in the curative setting. The late diagnosis in a considerable percentage of patients is directly related to aggressiveness, which is due to high tumour heterogeneity and the presence of resistant cells [2]. In fact, treatment success in the curative setting is frequently masked by resistance to radio- and/or chemotherapy, particularly CDDP. Importantly, the resistance phenotype can be reversed [3–5]. Therefore, therapeutic intervention against those mechanisms in resistant cells is urgently needed to sensitise HNSCC cells to CDDP treatment. 

Moreover, the following references has been added:

Nenclares P, Rullan A, Tam K, Dunn LA, St John M, Harrington KJ. Introducing Checkpoint Inhibitors Into the Curative Setting of Head and Neck Cancers: Lessons Learned, Future Considerations. Am Soc Clin Oncol Educ Book. 2022 Apr;42:1-16. doi: 10.1200/EDBK_351336. PMID: 35522916.

Ferris RL, Blumenschein G Jr, Fayette J, et al. Nivolumab for recurrent squamous-cell carcinoma of the head and neck. N Engl J Med. 2016;375:1856-1867

Burtness B, Harrington KJ, Greil R, et al.; KEYNOTE-048 Investigators. Pembrolizumab alone or with chemotherapy versus cetuximab with chemotherapy for recurrent or metastatic squamous cell carcinoma of the head and neck (KEYNOTE-048): a randomised, open-label, phase 3 study. Lancet. 2019;394:1915-1928.

  1. page 8 line 298; T refers to tumor and has nothing to do with metastatic lesions according to TNM, please explain.   

We thank the reviewer for this comment. Accordingly, the sentence has been modified to reflect the invasive and aggressive nature of primary tumors classified as T4. Associations of T4 category with metastatic setting have been eliminated.

Note that those patients with concomitant alterations in these 12 genes are mostly at the T4 category, a clinical parameter that refers to metastatic patients the most advanced and invasive type of primary tumors. 

  1. Please write the limitations of study (obviously small number of patients tissue).

This has been changed in the discussion section.

Reviewer 2 Report

The authors showed a novel therapeutic target in HNOSCC by cross-checking the RNA-seq results with the proteomic profiles. The research was demonstrated logically, and the manuscript was well-structured. The authors concluded that MAGEB2 is a crucial molecule in treatment resistance, especially CDDP, and might potentially become a new therapeutic target. I propose the following things which could improve the impact of this study.

How does MAGEB2 contribute to resistance to CDDP? What kind of signaling pathways are involved? What was the expression of the relevant molecules in the RNA-seq and the proteomic profiles? These should be described in the manuscript with appropriate references.

One of the other ways to support the conclusion is by checking the localization of MAGEB2 in the resected tissue or biopsy tissue by IHC of IF. Is there a significant difference between sensitive and resistance in the quantification of MAGEB2 expression?

Line 302: Figure 2 → Figure 3

Author Response

Reviewer 2

The authors showed a novel therapeutic target in HNSCC by cross-checking the RNA-seq results with the proteomic profiles. The research was demonstrated logically, and the manuscript was well-structured. The authors concluded that MAGEB2 is a crucial molecule in treatment resistance, especially CDDP, and might potentially become a new therapeutic target. I propose the following things which could improve the impact of this study.

How does MAGEB2 contribute to resistance to CDDP? What kind of signaling pathways are involved? What was the expression of the relevant molecules in the RNA-seq and the proteomic profiles? These should be described in the manuscript with appropriate references.

One of the other ways to support the conclusion is by checking the localization of MAGEB2 in the resected tissue or biopsy tissue by IHC of IF. Is there a significant difference between sensitive and resistance in the quantification of MAGEB2 expression?

The mechanisms by which MAGEB2 contributes to resistance are not the scope of this research. However, based upon MAGEB2 protein interactions (String) and literature search, we have hypothesized potential mechanisms by which MAGEB2 can contribute to the resistance phenotype including appropriate references.

The following paragraph has been included in the discussion section:

“According to the proteins that interact with MAGEB2, 50% of them belong to the cancer/testis antigen gene (GAGE) family. GAGE proteins are encoded by genes that are normally expressed only in the human germline, but are also expressed in various tumour types. We hypothesise that MAGEB2 could act at the level of two mechanisms of action largely mediated by the GAGE family: i) ability to create a more "open" chromatin structure by up-regulating histone 3 lysine 56 (H3K56Ac) acetylation, which could increase the efficiency of DNA repair, as described in a model of radioresistance in ovarian cancer (10.1016/j.celrep.2021 .109621) or ii) MAGEB2 could promote the development of immunosuppressive EMT (e.g. by down-regulating the expression of CXCL9 and CXCL10) and decreasing the recruitment of effector CD8+ T cells, thereby exerting resistance to checkpoint immunotherapy (10.1016/j.clim.2022.109045). Alternatively, MAGEB2 could induce stemness characteristics by modulating HDAC1 levels (10.1038/sj.bjc.6603163).”

Cui et al. Performed IHC of MAGEB2 and showed that MAGEB2 is located at the cytoplasm. We hypothesize that accordingly to the RNA-seq data, different MAGEB2 expression at protein levels would distinguish between resistant versus sensitive patients (10.1016/j.clim.2022.109045).

Line 302: Figure 2 → Figure 3

This has been changed accordingly.